# Provision and delivery of survivorship care for adult patients with haematological malignances: A scoping review protocol

Zoe-Anne V. Baldwin [1,2]*, Steph Busby [2,3], David Allsup [1,4], Judith Cohen [2,5], Olufikayo Bamidele [2,6]

1 Centre for Biomedicine, Hull York Medical School, University of Hull, Hull, United Kingdom, 2 Institute for Clinical and Applied Health Research, University of Hull, Hull, United Kingdom, 3 Academy of Primary Care, Hull York Medical School, University of Hull, Hull, United Kingdom, 4 Hull University Teaching Hospital Trust, Hull, United Kingdom, 5 Hull Health Trials Unit, Hull York Medical School, University of Hull, Hull, United Kingdom, 6 Cancer Awareness, Screening and Diagnostic Pathways Research Group, Hull York Medical School, University of Hull, Hull, United Kingdom

* z.baldwin-2021@hull.ac.uk

**Data Availability Statement:** No datasets were generated or analysed during the current study. All relevant data from this study will be made available upon study completion.

## Abstract

### Introduction

Haematological malignancies are a heterogenous group of blood and lymphatic cancers. Survivorship care is a similarly diverse term concerning patients' health and wellbeing from diagnosis to end of life. Survivorship care for patients with haematological malignances has traditionally been consultant-led and secondary care-based, although shifts away from this model have been occurring, largely via nurse-led clinics and interventions with some remote monitoring. However, there remains a lack of evidence regarding which model is most appropriate. Although previous reviews exist, patient populations, methodologies, and conclusions are varied, and further high-quality research and evaluation has been recommended.

### Aims

The aim of the scoping review this protocol describes is to summarise current evidence on the provision and delivery of survivorship care for adult patients diagnosed with a haematological malignancy, and to identify existing gaps to inform future research.

### Methodology

A scoping review will be carried out utilising Arksey and O'Malley's guidelines as its methodological framework. Studies published in the English language from December 2007 to the present will be searched on bibliographic databases, including Medline, CINAHL, PsycInfo, Web of Science, and Scopus. Papers' titles, abstracts, and full text will predominantly be screened by one reviewer with a second reviewer blind screening a proportion. Data will be extracted using a customised table developed in collaboration with the review team, and presented in tabular and narrative format, arranged thematically. Studies included will contain data regarding adult (25+) patients diagnosed with any haematological malignancy in

**Funding:** The authors received no specific funding for this work.

**Competing interests:** The authors have declared that no competing interests exist.

combination with aspects related to survivorship care. The survivorship care elements could be delivered by any provider within any setting, but should be delivered pre- or post-treatment, or to patients on a watchful waiting pathway.

## Registration

The scoping review protocol has been registered on the Open Science Framework (OSF) repository Registries (https://osf.io/rtfvq; DOI: 10.17605/OSF.IO/RTFVQ).

## Introduction

Haematological malignancies (HMs) are a heterogenous collection of cancers of the blood and lymphatic systems [1]. Under this umbrella term are five main categories: leukaemia, lymphoma, myeloma, myelodysplastic syndromes, and myeloproliferative neoplasms [2]. Within each category are numerous subtypes: these number greater than 100 in total, and collectively form the fifth most common cancer in the United Kingdom (UK) [3], after breast, prostate, lung, and bowel cancers [4]. Treatment modalities vary across the different pathologies, as do mortality rates [5]. As a consequence of research advances, survival outcomes have increased for many HMs [6]. However, lack of access to new therapies and delayed diagnoses continue to stall progress, especially for rarer cancers [7]. The features and needs for patients with HMs are unique and remain largely unmet [8].

When discussed in the context of cancer, survivorship is a broad term which relates to the wellbeing of an individual from diagnosis to the end of life [9]. Definitions vary and have evolved over time to include different aspects and timescales as research, treatment, and overall understanding have developed [10]. According to Armes et al. [11], the elements of survivorship care (SC) can be grouped under either physical or psychological headings. Physical issues relate to treatment consequences, follow-up for cancer recurrence or new primary malignancies, or aid with financial matters, including a return to employment. Psychological aspects could include support for patients to deal with fear and uncertainty, body image issues, or comorbid mental health conditions [11]. In addition, a key aim of survivorship should be to improve, promote, and support healthy lifestyles for patients diagnosed with cancer [12]. Furthermore, early support is essential to help improve patient outcomes [13]. Consideration should also be given to patients' available social and support networks as they too form a key part of the overall survivorship experience [14].

Until the release of the Department of Health's [15] Cancer Reform Strategy in 2007, healthcare policy in the UK largely viewed cancer as an acute disease. Since then, focus has shifted towards treating cancer as a long-term or chronic illness, with numerous multimodal initiatives conceived to help patients live with and beyond their initial diagnosis [16]. Multiple models of SC exist for patients with cancer [17]. The same is true for those diagnosed with a HM [18]. To help improve the quality of care for anyone affected by cancer, survivorship care plans (SCPs) incorporating both clinical and non-clinical elements have been introduced [17]. However, evidence which supports the use of SCPs plans in patients diagnosed with a HM is limited [19]. Similarly, in an attempt to move away from the traditional medical model of care, stratified care pathways were pilot tested across multiple tumour sites [20]. For lower risk patients, there was a shift towards supported self-management and remote monitoring after a short period of clinical follow-up, and all pathways investigated were deemed to be acceptable. Nevertheless, although some organisations within the UK have taken the guidance released

following pilot testing [21] and used it to help stratify pathways for certain HMs [22, 23], research into this area remains scarce.

Previous reviews of SC for adult patients with a HM have focused on either a specific cohort [24], a single aspect of survivorship such as work outcomes or unmet needs [25–27], study design [28], or single treatment modalities [29, 30]. A recent group of systematic reviews has been undertaken by researchers from the Fondazione Italiana Linfomi covering a variety of late effects and elements of survivorship. However, their patient population, inclusion criteria, and search strategies are narrower than the prospective review intends [31–36]. Other reviews that have specifically investigated models of SC have only included a small number of patients diagnosed with a HM alongside other tumour sites [37, 38], used less rigorous methodologies [8, 39], or need apprising with more recent research [18, 19].

There is currently no scoping review paper available which has summarised the breadth of existing research on this topic to help identify evidence and methodological gaps. Therefore, due to the large number of publications in the area (greater than 50 review-type papers alone), the disparity between previous research, and the varying methodologies, patient populations, and inclusion and exclusion criteria, a scoping review (ScR) to collate and discuss the existing research in this field is warranted. Because of the heterogeneity and range of available literature, a scoping review is an acceptable methodology to use rather than a full systematic review which requires a more focused research question and narrower selection of quality assessed papers [40].

## Aims

The aims of the proposed ScR are:

1. To summarise the current evidence on the provision and delivery of SC as it relates to adult patients diagnosed with a HM.

2. To identify existing gaps to inform future research in this area.

## Materials and methods

The ScR follows Arksey and O'Malley's [40] guidelines as a methodological framework. This guidance suggests that a ScR is an appropriate way to map a complex area of research as well as helping to illuminate any gaps in the existing literature. These disparities can then be addressed with further enquiry in the form of a systematic review or primary research study. Arksey and O'Malley describe five stages to consider when conducting a ScR, each of which will be discussed in greater detail below. To further aid with conduct and reporting, the review also follows the Preferred Reporting Items for Systematic reviews and Meta-Analyses extension for Scoping Reviews checklist (PRISMA-ScR) [41]. Additionally, a checklist for the development and reporting of this ScR protocol [42] is included within the supplementary material (S1 Checklist), along with a proposed timeline for completion (S1 Timeline).

### Stage 1: Identifying the research questions

**Research question.**   What models of SC are currently provided for adult patients with a HM and how are they delivered?

To answer this, the following supplementary questions will also be investigated:

1. What does SC look like for adult patients with HMs: who are the stakeholders, when and where is care delivered, and for how long?

2. Does the approach to survivorship alter for different subtypes of HM, and in different geographical locations?

3. How do patients, caregivers, and clinicians perceive SC as it is currently being delivered?

The research question has been developed using a PCC (Population, Concept, Context) framework as recommended by the Joanna Briggs Institute [43] for ScRs:

**Population:** Adult patients with a HM.

**Concept:** Models of SC.

**Context:** Although SC could be delivered in any location, by any healthcare provider, and following any treatment modality regardless of intent, patients should have finished active treatment and be moving, or have already moved, onto follow-up. We will also include studies which report on patients on a watchful waiting pathway or any pre-treatment SC interventions.

## Stage 2: Identifying relevant studies

**Data sources.** Searches for published studies, regardless of methodology, will be carried out on the following databases: MEDLINE, Embase, CINAHL, the Cochrane Library (Cochrane Central Register of Controlled Trials and the Cochrane Database of Systematic Reviews), PsycInfo, and Scopus. To complement the database searches, grey literature will be identified on Web of Science, Open Access Theses and Dissertations, the World Health Organization International Clinical Trials Registry Platform, and the International Standard Randomised Controlled Trial Number registry. Additionally, websites of guideline collections, development agencies, and professional societies will be searched for relevant practice guidelines, such as the National Institute for Health and Care Excellence (NICE) and National Guideline Clearinghouse summaries archive. Key charity websites for HMs will also be scrutinised for appropriate literature, as well as forward and backward citation searching of included studies and review papers, contacting authors, the use of existing networks, and peer discussion. Searches will be re-run prior to final data collation to include any newly published, suitable articles.

**Search strategy.** The ScR search strategy was systematically developed in consultation with an experienced subject librarian and using an iterative process. Initial, exploratory searches using keywords and indexed terms (where appropriate) broadly related to HM, survivorship, and the various aspects of SC will be performed on the above databases. The search terms will then be refined or expanded as necessary to ensure the search is comprehensive. The MEDLINE search strategy will also be peer reviewed by an information specialist to check for optimisation. An example search strategy for MEDLINE can be found in the supplementary materials of this protocol (S1 File). This search strategy will be adapted to each database as required.

Searches will be limited to English language only due to limitations in time and resources available to translate non-English studies. We will further limit the searches to papers published from December 2007 to the current date. This date range has been chosen as it coincides with the publication of the Cancer Reform Strategy [15] which signified a significant policy shift away from treating cancer as an acute condition and starting to view it as a long-term health condition with more of a focus on SC.

Searches will also be limited to papers concerning adult (25+) humans. In this review, we have chosen not to include patients classified as teen and young adult (TYA), i.e., those aged 16–24, as TYA patients have different needs to those of the remaining adult population [44]. Because of this, in the UK, their treatment and ongoing care falls under specific remits and

specialised services until they reach the age of 25 [45]. The TYA population have also been widely investigated in relation to their experiences of, and issues surrounding, SC and long-term follow-up [46–48]. Therefore, TYA patients have been excluded from the current review to focus on the adult and elderly population. Traditionally, adult patients diagnosed with a HM feature in much smaller numbers within SC research compared to patients with other tumour types [18, 26]. Similarly, despite more elderly patients being diagnosed with HMs than other age groups, they have previously been largely excluded from clinical trials [49, 50]. It is therefore proposed that by concentrating on patients diagnosed with a HM over the age of 25, the review will provide a positive contribution to this under studied population.

**Inclusion and exclusion criteria.**   The PCC framework has helped inform the inclusion and exclusion criteria for which studies should be included in the ScR (Table 1).

All published articles meeting the above criteria will be considered regardless of study design, including systematic or other review papers, as well as grey literature such as government papers, clinical guidelines, or conference proceedings. However, articles which lack empirical data such as opinion pieces, case studies, commentaries, letters to the editor, or protocols will be excluded. Unpublished data (for example, in thesis or conference abstracts) that has later appeared in a peer-reviewed journal will also be excluded in preference of the published data, and any divided publication papers which share data sets will only be included once. Relevant papers where the full text is unavailable will also be excluded, although every effort will be made to source these, such as contacting the authors or requesting via the library.

**Table 1. PCC framework.**

| Population | Inclusion criteria:<br>• Adult (25+) humans with a diagnosis of a haematological malignancy (e.g., leukaemia, lymphoma, myeloma, myelodysplasia, or myeloproliferative neoplasm).<br>• We will also include papers concerning caregivers, either in combination with adults with a haematological malignancy diagnosis or as a standalone.<br>Exclusion criteria:<br>• Patients with a haematological malignancy who are <25 years old.<br>• Patients with another form of malignancy other than the haematological subtypes noted above.<br>• However, papers which detail both haematological and other forms of malignancy, or other morbidity, will be included provided there is a clear separation of data.<br>• Papers which discuss adults (25+) and teens (16–18) or young adults (19–24) diagnosed with a haematological malignancy will also be included provided there is a clear separation of data. |
|---|---|
| Concept | Inclusion criteria:<br>Any single or combined aspect of survivorship care (e.g., monitoring and managing late effects of treatment; prevention and detection of new primaries; health promotion and psychological wellbeing; monitoring for recurrences).<br>Exclusion criteria:<br>Papers whose primary focus is the efficacy, prognostic value, or overall survival of a treatment or treatments (including pharmaceutical or surgical interventions and other treatment regimens), or for specific subtypes of haematological malignancy in general. |
| Context | Inclusion criteria:<br>• Survivorship care could be delivered by any provider (e.g., nurse, consultant, general practitioner, or allied health professional), in any setting (e.g., hospital, primary care, community), or geographical location (though primary interest is UK-based data), following any treatment intent (i.e., palliative or curative).<br>• We will also consider papers where patients are on a watch and wait pathway, or any papers concerning pre-habilitation (e.g., physical, psychological, or other holistic support pre-treatment).<br>Exclusion criteria:<br>• Patients still on active treatment of any modality, and which discuss the immediate consequences and management of these treatments.<br>• Papers concerning patients receiving end of life care (i.e., patients deemed to be within the last few months of life). |

**Table 2. Data extraction table.**

| Authors, Year & Country | Aims & Objectives | Survivorship Definition | Study Design & Methodology | Participant Characteristics | Treatment Details | Study/ Model Setting | Provider of Model/ Intervention | Other Features of Model/ Intervention | Key Findings & Recommendations |
|---|---|---|---|---|---|---|---|---|---|
|  |  |  |  |  |  |  |  |  |  |

## Stage 3: Study selection

Search results from the various databases will be imported into EndNote [51] and duplicates removed. To aid in the screening and data extraction process and for ease of collaboration, the remaining references will be uploaded into Covidence [52]. Study selection will mainly be carried out by one reviewer, initially by reading study titles and abstracts, then by assessing the full text of the articles against the inclusion and exclusion criteria. To enhance rigour, a proportion of studies will also be screened by a second reviewer. Regular meetings will be held with the review team throughout this process to ensure the agreed eligibility criteria are being followed and papers are being appraised appropriately. Any conflicts will be resolved by discussion with a third member of the review team. The study selection process will be reported using a PRISMA-ScR flow diagram [41].

## Stage 4: Charting the data

Data extraction will mainly be undertaken by one reviewer using a standardised proforma which will be developed following discussions with the review team. To enhance rigour, data extraction of a proportion of the studies selected for inclusion will also be carried out by a second reviewer. A third reviewer from the team will be available to mediate any unresolved conflict.

Types of data to be extracted will include:

- Authors, year, and country

- Aims and objectives, if appropriate

- Description or definition of survivorship care utilised, if available

- Study design and methodology, if appropriate

- Characteristics of participants (e.g., number, demographics, subtype of haematological malignancy)

- Details of any treatment received or proposed (e.g., pre/post-treatment, when finished, modalities, length, intent)

- Study or model setting (e.g., hospital, community, primary care)

- Provider of model or intervention (e.g., consultant, nurse, general practitioner)

- Other characteristics of model or intervention (e.g., element/s of survivorship care considered, length and type of intervention, any onward referrals or involvement of external agencies)

- Key findings and recommendations

A draft data extraction table has been presented below (Table 2).

### Stage 5: Collating, summarising, and reporting the results

The results will be presented visually and textually, using tables to display the data charted from the various sources, and via a narrative summary. Themes will be extracted from the data and articles grouped together within them (e.g., literature discussing similar models or aspects of survivorship care, as well as sources from similar geographical locations). The component parts of these themes will be discussed individually, as they relate to one another, gaps in literature, and recommendations for future research. Data collation, summarisation and reporting will be carried out by one reviewer supported by consultations and evaluations with and by the wider review team. Any conflicts will be resolved by discussion.

## Discussion

### Dissemination

Review findings will be disseminated via oral presentations to key stakeholders, including members of the healthcare research community, patients, and clinical staff, as well as through publication in a scientific peer reviewed journal. The aim of any potential engagement events to discuss the results of the scoping review would be to facilitate both patient and public involvement (PPI) workshops and further discussions with clinicians. These dialogues will help provide useful feedback which will be used to inform the design of a future research project. The proposed project intends to develop a survivorship care intervention for UK-based adult patients with a HM. Data from PPI workshops and clinical consultations will be combined with the scoping review data and used to inform the design of the study protocol for this research project.

### Registration

A project has been created for the ScR on the OSF repository and the protocol and search strategies uploaded. The project has also been registered with the OSF Registries.

- OSF project: https://osf.io/rtfvq

- DOI: 10.17605/OSF.IO/RTFVQ

## Supporting information

**S1 Checklist. PRISMA-P checklist.**
(DOCX)

**S1 Timeline. Proposed timeline for completion.**
(DOCX)

**S1 File. Sample search strategy for Ovid MEDLINE.**
(DOCX)

## Acknowledgments

The authors would like to acknowledge Academic Library Specialists Fiona Ware and Sara Hastings (Brynmor Jones Library, University of Hull) who assisted in developing a search strategy for this ScR, and Information Specialist Sarah Greenley (Cancer Awareness, Screening and Diagnostic Pathways Research Group, Hull York Medical School, University of Hull) who peer-reviewed the draft MEDLINE search strategy.

## Author Contributions

**Conceptualization:** Zoe-Anne V. Baldwin, David Allsup, Judith Cohen, Olufikayo Bamidele.

**Data curation:** Zoe-Anne V. Baldwin.

**Investigation:** Zoe-Anne V. Baldwin, Steph Busby.

**Methodology:** Zoe-Anne V. Baldwin, David Allsup, Judith Cohen, Olufikayo Bamidele.

**Project administration:** Zoe-Anne V. Baldwin.

**Resources:** Zoe-Anne V. Baldwin.

**Supervision:** David Allsup, Judith Cohen, Olufikayo Bamidele.

**Validation:** David Allsup, Judith Cohen, Olufikayo Bamidele.

**Visualization:** Zoe-Anne V. Baldwin.

**Writing – original draft:** Zoe-Anne V. Baldwin.

**Writing – review & editing:** Steph Busby, David Allsup, Judith Cohen, Olufikayo Bamidele.

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
