## [Decision Letter · Decision Letter 0]

16 Feb 2023

Provision and delivery of survivorship care for adult patients with haematological malignancies: a scoping review protocol

PONE-D-22-34931

Dear Dr. Baldwin,

We’re pleased to inform you that your manuscript has been judged scientifically suitable for publication and will be formally accepted for publication once it meets all outstanding technical requirements.

Kind regards,

Academic Editor

PLOS ONE

Additional Editor Comments (optional):

Reviewers' comments:

Reviewer's Responses to Questions

**Comments to the Author**

1. Does the manuscript provide a valid rationale for the proposed study, with clearly identified and justified research questions?

Reviewer #1: Yes

Reviewer #2: Yes

2. Is the protocol technically sound and planned in a manner that will lead to a meaningful outcome and allow testing the stated hypotheses?

Reviewer #1: Yes

Reviewer #2: Yes

3. Is the methodology feasible and described in sufficient detail to allow the work to be replicable?

Reviewer #1: Yes

Reviewer #2: Yes

4. Have the authors described where all data underlying the findings will be made available when the study is complete?

Reviewer #1: Yes

Reviewer #2: Yes

5. Is the manuscript presented in an intelligible fashion and written in standard English?

Reviewer #1: Yes

Reviewer #2: Yes

6. Review Comments to the Author

You may also provide optional suggestions and comments to authors that they might find helpful in planning their study.

Reviewer #1: This is good study which will focus to solve the future treatment and policy plan for the haematological disorder. all the aim and objectives are well addressed and i am recommending this article for publication.

Reviewer #2: The manuscript by Baldwin et al. ‘Provision and delivery of survivorship care for adult patients with haematological malignancies: a scoping review protocol’ is very interesting. This study protocol aims to summarize current evidences on the provision and delivery of survivorship care for adult patients diagnosed with a hematological disorders and to identify existing gaps to improve future investigations. The manuscript is well written and provides enough strategies to fulfill aims of the study protocol, though language may need some polishing for better flow. In conclusion, I support publication of this study protocol and sure that it will be interesting for researchers working in the hematology field.

7. PLOS authors have the option to publish the peer review history of their article (what does this mean?). If published, this will include your full peer review and any attached files.

Reviewer #1: **Yes: **Vivek Singh

Reviewer #2: No

---

## [Editor Report · Acceptance letter]

21 Feb 2023

PONE-D-22-34931 

Provision and delivery of survivorship care for adult patients with haematological malignancies: a scoping review protocol 

Dear Dr. Baldwin:

I'm pleased to inform you that your manuscript has been deemed suitable for publication in PLOS ONE. Congratulations! Your manuscript is now with our production department. 

Kind regards, 

on behalf of

Dr. Robert Jeenchen Chen 

Academic Editor

PLOS ONE